# Quantum Applications of an Atomic Ensemble Inside a Laser Cavity

Andrei Ben Amar Baranga, Gennady A. Koganov, David Levron, Gabriel Bialolenker and Reuben Shuker *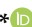

Physics Department, Ben Gurion University of the Negev, Beer Sheva 84105, Israel; andreib@bgu.ac.il (A.B.A.B.); quant@bgu.ac.il (G.A.K.); dlevron@bgu.ac.il (D.L.)
* Correspondence: shuker@bgu.ac.il

**Abstract:** Many quantum device signals are proportional to the number of the participating atoms that take part in the detection devices. Among these are optical magnetometers, atomic clocks, quantum communications and atom interferometers. One way to enhance the signal-to-noise ratio is to introduce atom entanglement that increases the signal in a super-radiant-like effect. A coherent em field inside a laser cavity is suggested to achieve atoms' correlation/entanglement. This may also play an important role in the basic quantum arena of many-body physics. An initial novel experiment to test the realization of atoms' correlation is described here. A Cs optical magnetometer is used as a tool to test the operation of a cell-in-cavity laser and its characteristics. A vapor cell is inserted into an elongated external cavity of the pump laser in Littrow configuration. Higher atom polarization and reduced laser linewidth are obtained leading to better magnetometer sensitivity and signal-to-noise ratio. The Larmor frequency changes of the Free Induction Decay of optically pumped Cs atomic polarization in the ambient earth magnetic field at room temperature is measured. Temporal changes in the magnetic field of less than 10 pT/$\sqrt{\text{Hz}}$ are measured. The first-order dependence of the magnetic field on temperature and temperature gradients is eliminated, important in many practical applications. Single and gradiometric magnetometer configurations are presented.

**Keywords:** sensing; coherent photon states; quantum correlations

## 1. Introduction

Many quantum device signals are proportional to the number of the participating atoms N, that take part in the detection devices. Among these are optical magnetometers [1], atomic clocks [2], quantum communications [3,4] and atom interferometers [5]. One way to enhance the related signal-to-noise ratio is to introduce atom entanglement that results in an increase of the signal in a super-radiant-like effect. The resulting signal is then proportional to $N^2$ rather than N. A coherent em field inside a laser cavity is suggested to achieve atoms' correlation/entanglement. This may also play an important role in the basic quantum arena of many-body physics [6]. In this article, we introduce an attempt to demonstrate such an effect. The idea is to expose the atoms to a coherent em field inside a laser cavity. The aim is to show lasing at resonance while there exists resonance absorption in its cavity. The idea is that atoms correlate due to the coherence of the em field. We have recently shown theoretically [7] the collective behavior of the atoms such as the dressing of quantum states by the atoms and the $N^2$ dependence of the absorption coefficients. In a brief review, we describe the basics of the theory. The collective Hamiltonian and the derived density matrix show the many-body collective absorption coefficient scaling as $N^2$. It is shown that the collective effect depends on the number of the relevant photon, namely, a strong coherent field is imperative. Such a requisite is quite demanding. To fulfill such a requirement, we introduced the atoms inside the laser cavity. The initial operation of a coherent field exposing the atoms to such a field, even feeble, is required. Such a successful attempt is described in this paper. In our experiment, a hot vapor cell is employed and the detector device is an optical magnetometer.

An experimental test of the atoms' correlation effect toward an improvement of the signal-to-noise ratio is performed utilizing the performance of an optical magnetometer. Optical magnetometers have drawn interest and many studies due to their importance in basic and applied research. The main effort is devoted to the well-known magnetically shielded Spin Exchange Relaxation Free (SERF) magnetometer [8–10]. In this magnetometer, optical pumping is used to polarize alkali atoms with a circularly polarized laser beam and any change in the magnetic field is probed by a linearly polarized beam. Over the last ten years, less sensitive but unshielded magnetometers, based on the Bell–Bloom method (cw lasers) or based on measuring the Larmor frequency changes of the Free Induction Decay (FID) of the atomic polarization indued by pulsed lasers, have been developed. These devices open the way for many applications requiring a sensitivity of several pico-Tesla (pT)/$\sqrt{Hz}$ at 1 Hz in measuring minute changes in the ambient earth magnetic field [11–13]. The latter, called "pulsed magnetometer", offers a direct measurement of the Larmor frequency, independent to the first order on temperature and temperature gradients over the cell, an important and favorable feature in many practical applications. This method also avoids possible complications with the Electromagnetically Induced Transparency process (EIT) when using polarization intensity changes as a method for magnetometry [14]. Just recently, an unshielded, portable gradiometer with a sensitivity of 16 fT/cm/$\sqrt{Hz}$ in open space with reduced size and lower power consumption based on multipass cells pulsed magnetometers has been demonstrated [15,16].

In the present paper, we present a multipass methodology based on introducing the vapor cell inside the pump-laser Littrow configuration external cavity. This configuration works with AR-coated and with uncoated diode lasers. It provides a higher and coherent polarizing field, quantum coherence of the em field inside a cavity and possible atoms' correlation [7].

## 2. Theoretical

Recently, we have introduced a generic collective excitation exchange operator that describes the photon-assisted excitation exchange, real or virtual, between two atoms and conserves the total number of excitations in the joint electromagnetic field and atoms' quantum system [7]. This novel operator exhibits photon-assisted atom correlation. It results in a new dressing of the atomic levels, i.e., dressing by the number of active atoms $N$. It also brings about $N^2$ dependence in atomic parameters, such as absorption, a signature of achieving collective behavior. Another virtue of atom correlations is that they may provide the possibility of achieving the Heisenberg limit of quantum noise $1/N$, rather than the standard quantum noise limit of $1/\sqrt{N}$. The $N$-dressing process was observed in [17]. Our theory generalizes their calculations and effectively accounts for their experimental data [18].

We briefly outline the results presented in [7]. For a system of N two-level atoms interacting with a quantized single mode of electromagnetic field inside a resonant cavity, the interaction Hamiltonian (in units of $\hbar = 1$) reads:

$$\hat{H} = \sum_{i=1}^{N} (g\hat{\sigma}_i \hat{a}^\dagger + H.c.) + \Omega_c \sum_{i \neq j} \hat{a}^\dagger \hat{\sigma}_i \hat{\sigma}_j^\dagger \hat{a} \tag{1}$$

Here, $g$ is a coupling constant, $\hat{a}^\dagger$ and $\hat{a}$ are operators of creation and annihilation of photons, $\hat{\sigma}_i = |a\rangle_{ii}\langle b|$ is the lowering operator of $i$-th atom. The first term is a standard interaction Hamiltonian for a two-level system driven by a resonant laser field. The second term is the novel operator describing the photon-induced correlation between different atoms introduced in [7], $\Omega_c$ stands for inter-atomic correlation strength. Eigenvalues of the Hamiltonian (1) are:

$$E = \frac{1}{2}N\left[\Omega_c(N-1)n \pm \sqrt{4ng^2 + \Omega_c^2 n^2(N-1)^2}\right] \tag{2}$$

where $n = \langle \hat{a}^\dagger \hat{a} \rangle$ is the number of photons in the cavity. These eigenvalues manifest a collective characteristic seen in the asymptotic $N^2$ dependence at large values of $N$:

$$E \approx \Omega_c n N^2 \tag{3}$$

Similar collectivity is also manifested in the behavior of the absorption coefficient. We solve semiclassical equations for the atomic density matrix derived from the Hamiltonian (1) and calculate the absorption coefficient $\alpha$ contained in the equation for the number of photons $dn/dx = -\alpha(\Omega_c)n/c$ [7]:

$$\alpha(\Omega_c) = \frac{4g^2 N \gamma}{\gamma^2 + 8g^2 n + 4n^2\Omega_c^2(N-1)^2} \tag{4}$$

where $g$ is spontaneous decay rate. In Figure 1, the absorption contained $a$ is plotted as a function of the number of atoms. This is a clear indication of the effect of the correlation on the absorption coefficient and of the collective signature of a large number of atoms. The ratio of the absorption coefficient with correlations $a(W_c)$ to $a(0)$ of a large number of atoms and under strong field assumption $\gamma^2 \ll ng^2$ scales is $1/N^2$:

$$\frac{\alpha(\Omega_c)}{\alpha(0)} \approx \frac{2g^2}{n\Omega_c^2 N^2} \tag{5}$$

The atomic correlations inhibit the absorption by a factor of $N^2$ and consequently enhance the signal by a factor of $N^2$.

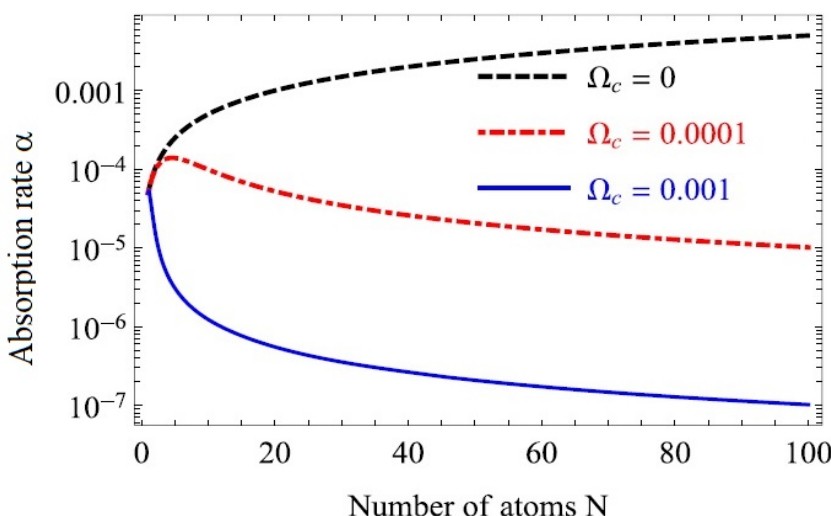

**Figure 1.** Absorption coefficient $\alpha$ as a function of the number of atoms N for no correlation $\Omega_c = 0$ (black dashed line), in presence of correlation $\Omega_c = 0.0001$ (red dash-dotted line) and $\Omega_c = 0.001$ (blue solid line).

## 3. Experimental

A Cs vapor cell is introduced in the external cavity of a diode laser operated in Littrow–Melbourne configuration [19], as Figure 2 illustrates. The output laser beam is spectrally narrowed and tuned to resonance. The setup includes a CW, AR-coated diode laser at Cs D2 line @852 nm (Toptica LD-0860-0080-AR-1, 90 mW, TOPTICA Photonics AG, Lochhamer Schlag 19, 82166 Graefelfing (Munich), Germany), an f11 aspheric lens, a cylindrical Cs vapor cell of 25 mm diameter and 50 mm length lens containing 5 Torr of Nitrogen, 7.5 torr Ne and a drop of Cs (Sacher Lasertechnik, LLC 5765, Equador Way, Buena Park, CA 90620, USA), an 1800 grooves/mm holographic diffraction grating and a folding mirror. The cavity length is approximately 30 cm long. The extended external cavity reduces the

laser linewidth from several hundred GHz to several tens of MHz (See Appendix A). As expected, the longer the cavity, the smaller the linewidth [20]. The fluorescence in the relatively low-pressure vapor cell is an indication of the tuned wavelength to resonance. As one can see in the figure, the original laser radiation of ~2 nm at FWHM is reduced to less than 1 pm as measured by a HighFinesse WS-6 wavemeter (HighFinesse GmbH, Wöhrdstraße 4, 72072 Tübingen, Germany). The diode laser linear polarization is vertical and so are the grating grooves. In this specific measurement, the diode laser is AR coated and the vapor cell is uncoated, but a similar operation is observed with uncoated diodes, although less efficient. A similar operation of a pulsed diode laser with a vapor cell inside the external cavity was demonstrated too.

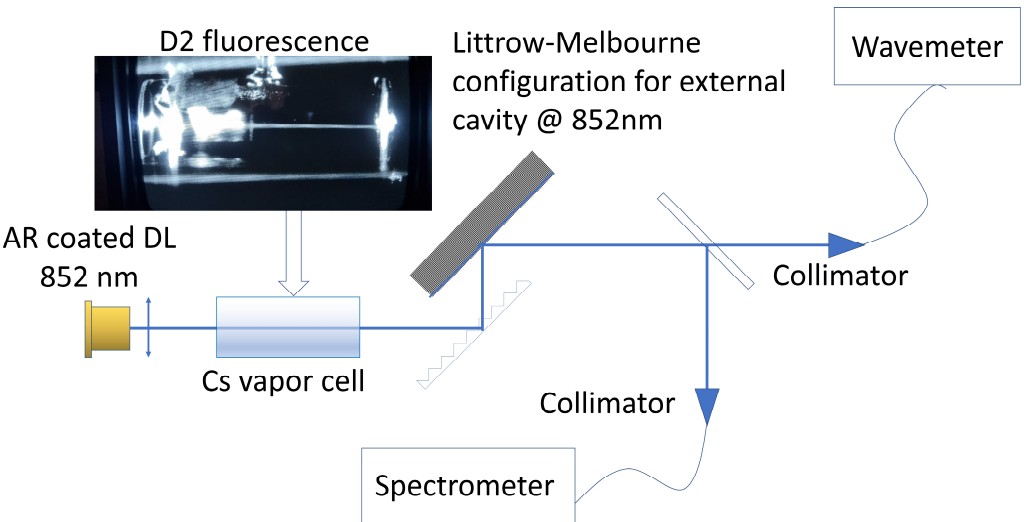

**Figure 2.** Diode laser with a Cs vapor cell inside the elongated external cavity. Shown is the fluorescence through the cell at resonant D2 line.

The heart of a pulsed magnetometer is the employment of a short-pulse pump laser (less than 200 ns pulse duration) followed by the Free Induction Decay of the Cs atom polarization pulse [21]. To increase optical pumping efficiency, we introduced an un-coated Cs vapor cell inside the pulsed pumping diode laser external cavity in a configuration, as Figure 2 shows.

For the pulsed magnetometer, a multimode, broadband (~100 GHz), high-power pulsed laser diode, Intense 2150-9 mm-89505 (Intense, 45 Horsehill Road, Suite 107B, Cedar Knolls, NJ 07927, USA), vertically polarized, is used with the elongated external cavity and in-cavity vapor Cs cell detailed before. The laser is resonant to Cs D1 line at 895 nm, operated at a 1–5 kHz repetition rate of 100–300 ns pulses. The pulse duration is limited by the manufacturer and was not optimized. The peak pump laser power during the pulse is several watts.

A $\lambda/4$ waveplate, WP1, is added inside the cavity to circularly polarize the laser entering the vapor cell to optically pump the atoms. A second $\lambda/4$ waveplate, WP2, with its axis parallel to the first one, is added after the cell to turn the circular polarization into a horizontal linear polarization, preferable for S first-order reflection from the vertically grooved Littrow grating [22]. Figure 3 shows this setup. An analysis of the pump laser polarization through the cavity is conducted by Jones matrices [23]. The result of the calculation, positioning the plates WP1 and WP2 such that their main axes are relatively parallel, is that the atomic polarization increases with each passage of the laser beam in the cell for both directions of beam propagation. As Figure 4 shows, the circular polarization of the reflected beam inside the cell is in the same direction as the original beam, increasing atom polarization and, accordingly, the magnetometer signal.

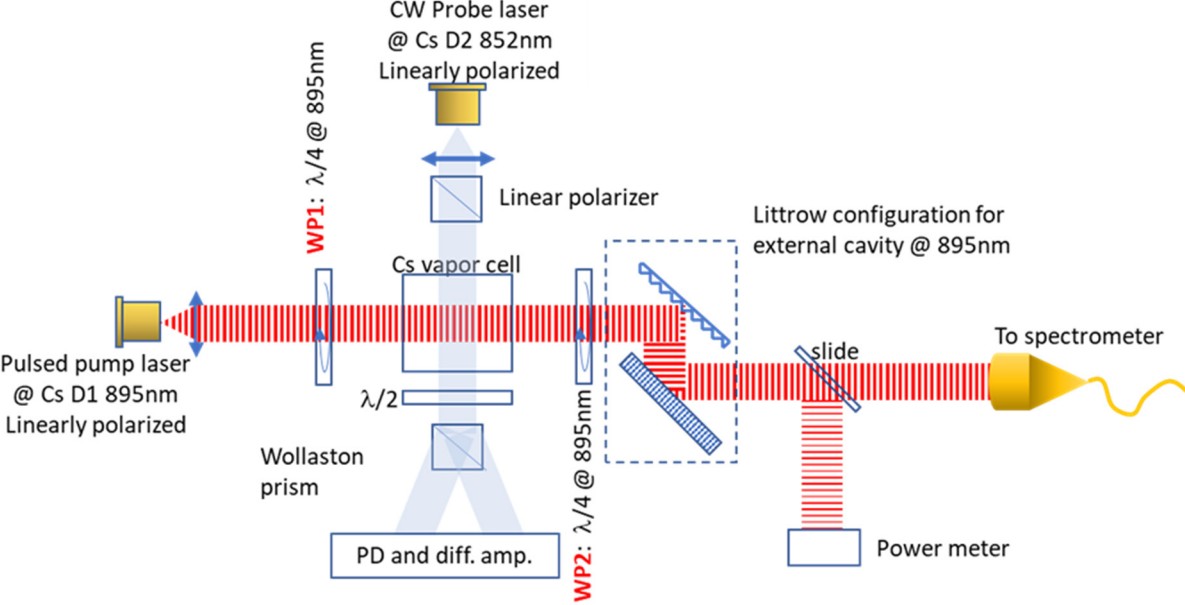

**Figure 3.** Pulsed magnetometer with the vapor cell inside the pump laser cavity. The probe laser is perpendicular to the pump.

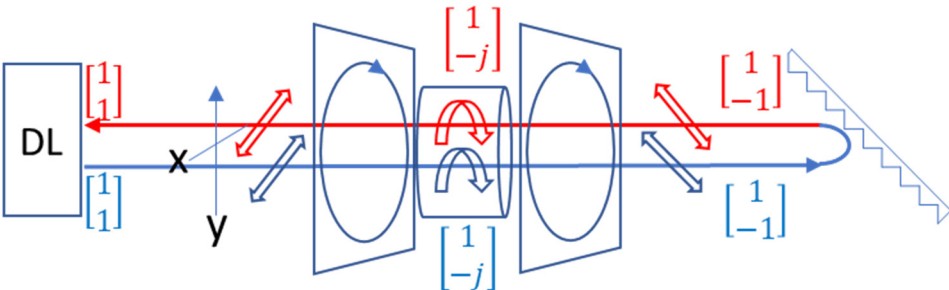

**Figure 4.** Jones matrix presentation for laser beam polarization inside the Littrow external cavity.

The transverse polarization of the Cesium atoms is measured by detecting the optical rotation induced by the paramagnetic Faraday rotation of the linearly polarized probe light, propagating through the atomic vapor perpendicular to the pump beam, as Figure 3. We use a balanced polarimeter setup comprising a Wollaston prism and a Thorlabs' PDB210A—Large-Area Balanced Photodetector. The probe laser is a Toptica LD-0852-015 DFB-1 CW, low power (<15 mW), narrow bandwidth distributed feedback laser diode (DFB) linearly polarized, detuned by approximately 10 pm from the Cs D2 line at 852 nm; furthermore, linearly polarized by a Glan-Laser polarizer for a higher contrast ratio. The probe power and detuning have been optimized for the highest signal as well as the two $\lambda/4$ waveplates' orientation.

The acquired raw data consist of a series of FID waveforms at pump pulse repetition rate, with a period T (e.g., one millisecond at 1 kHz). The functional form of the waveform ($W_f$), as Equation (6) shows, is obtained from solving the Bloch equation for polarization.

$$W_f = A \cdot \exp(-\Gamma t) \cdot \sin(2\pi f t) \tag{6}$$

The decay constant G is the reciprocal of the sum of all decay rates of the polarization, and the oscillation frequency $f$ is the Larmor frequency of the ground level of the cesium atom. Assuming constant frequency for the waveform in the $i$th period, $(I-1)T < t < iT$ for $i = 1$ to N (N is the total number of pulses per measurement), we calculate the Larmor frequency fi (proportional to the magnetic field) by either zero crossing or Interpolated Fourier Transform (FFT) algorithms. Defining the function $F(t) = fi$-average($f_i$) for $(i-1)$ $T < t < iT$, for $i = 1$ to $N$, we convert it to magnetic field values $B(t)$ using the gyromagnetic

ratio of Cs, and calculate its Power Spectral Density (PSD) and Amplitude Spectral Density (ASD) in the frequency domain:

$$PSD_{B(t)}(\omega) = 2 \cdot \frac{T}{N} \cdot abs(FFT_{B(t)}(\omega))^2 \qquad (7)$$

for $\omega = (1, \ldots, N/2)/T$; dimensionality: Tesla$^2$/Hz.

$$ASD_{B(t)}(\omega) = \sqrt{PSD_{B(t)}(\omega)} \qquad (8)$$

dimensionality: Tesla/$\sqrt{\text{Hz}}$.

The noise floor of the measurement is the ASD outside the signal peaks. The intensity of the applied magnetic field calculated from the spectral density curves is the square root of the integral of B2 obtained by integrating the PSD over the peak. The maximum bandwidth of our pulsed magnetometer operated at 1 kHz is 500 Hz (Nyquist frequency) but it is sensitive to much higher frequencies. However, its sensitivity decays with frequency. The lowest measured frequency is 0.01 Hz.

Figure 5 compares ASD spectra measured in two experiments: one with the cell inside the laser cavity and one with the cell outside the cavity at room temperature of 25 °C. The external magnetic fields of 1.8 nT at 13 Hz and 11.3 Hz were applied on the cell inside the cavity as well as on the cell outside the cavity, respectively. This applied field is depicted on the background of the ambient earth's magnetic field of tens of mT. One can see that the noise floor is lowered by introducing the cell inside the cavity without affecting the applied field intensity measurement. Other peaks appearing in the spectra are at 50 Hz and its harmonics existing in the laboratory.

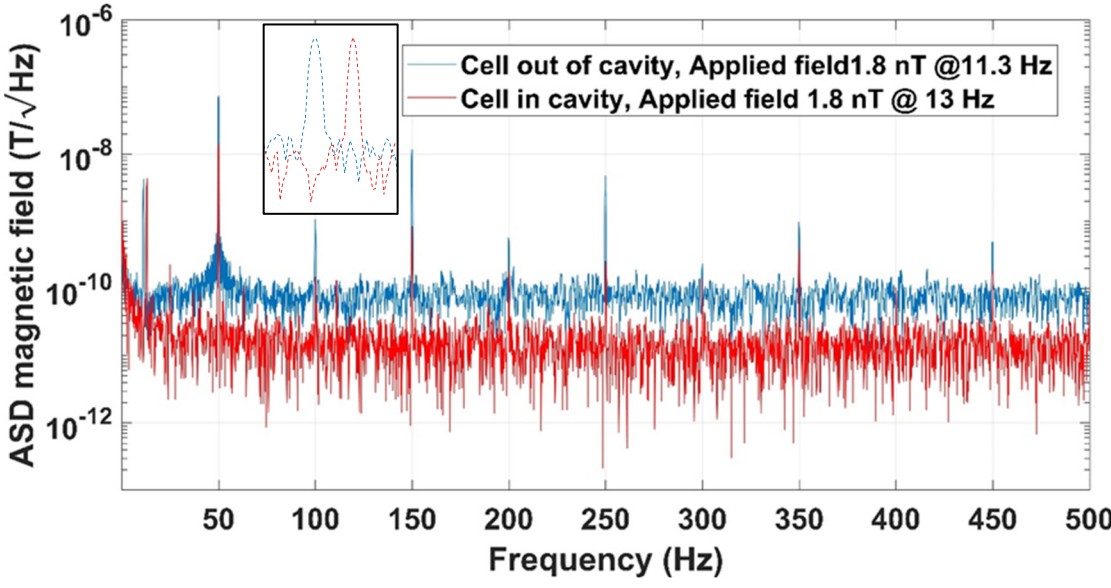

**Figure 5.** ASD of the pulsed magnetometer with the vapor cell inside the pump laser cavity and outside the cavity. The noise floor is lowered by introducing the cell inside the cavity without affecting the applied field intensity measurement as emphasized in the insert. The insert shows the field at 11.3 Hz and 13 Hz.

A similar magnetometer configuration is used for gradiometric measurements. For a small baseline magnetometer, the probe beam is split into two equal and parallel beams depicting atom polarization by crossing a single vapor cell through two separate lines separated by a short distance, as Figure 6A shows. For the large baseline gradiometer, the probe beam split into two, crossing two separate cells, with the two of them located inside the external cavity of the pump laser at 10–15 cm distance, as Figure 6B shows.

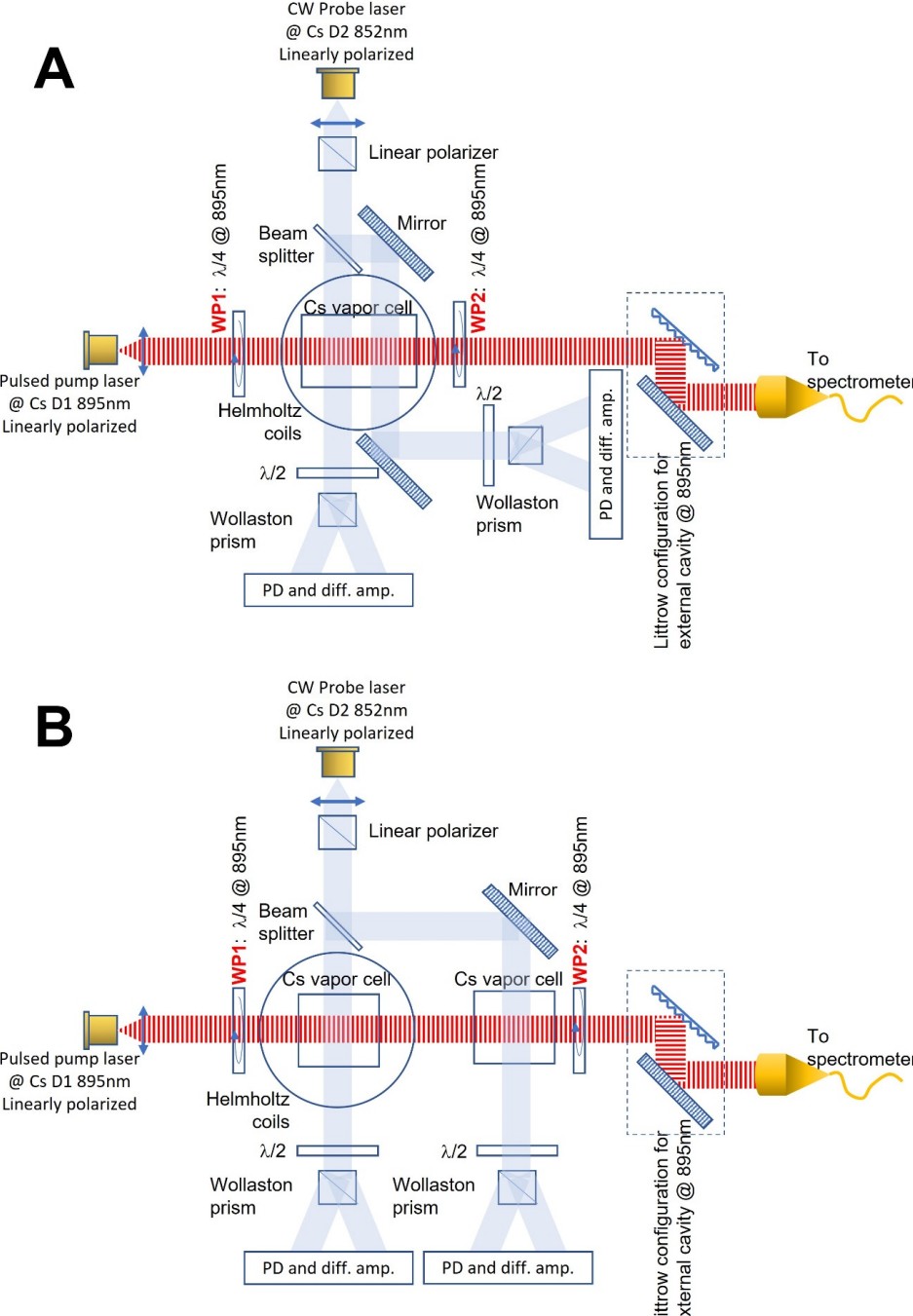

**Figure 6.** (**A**) Small baseline gradiometer, single vapor cell. (**B**) Large baseline gradiometer with two vapor cells inside the pump laser cavity.

An amplitude spectral density (ASD) of $10\ \mathrm{pT}/\sqrt{\mathrm{Hz}}$ of a single magnetometer at frequencies higher than 10 Hz was measured in the laboratory at room temperature. ASD of less than $10\ \mathrm{pT}/\mathrm{mm}/\sqrt{\mathrm{Hz}}$ at 1 Hz has been demonstrated in the laboratory in gradiometric configuration with a 1.5 mm baseline, operated at 1 kHz of 200 ns pulses and 8–100 s of data acquisition time. Figure 7 shows the results of such a gradiometric measurement. Better sensitivity is expected outdoors, far from buildings' noise.

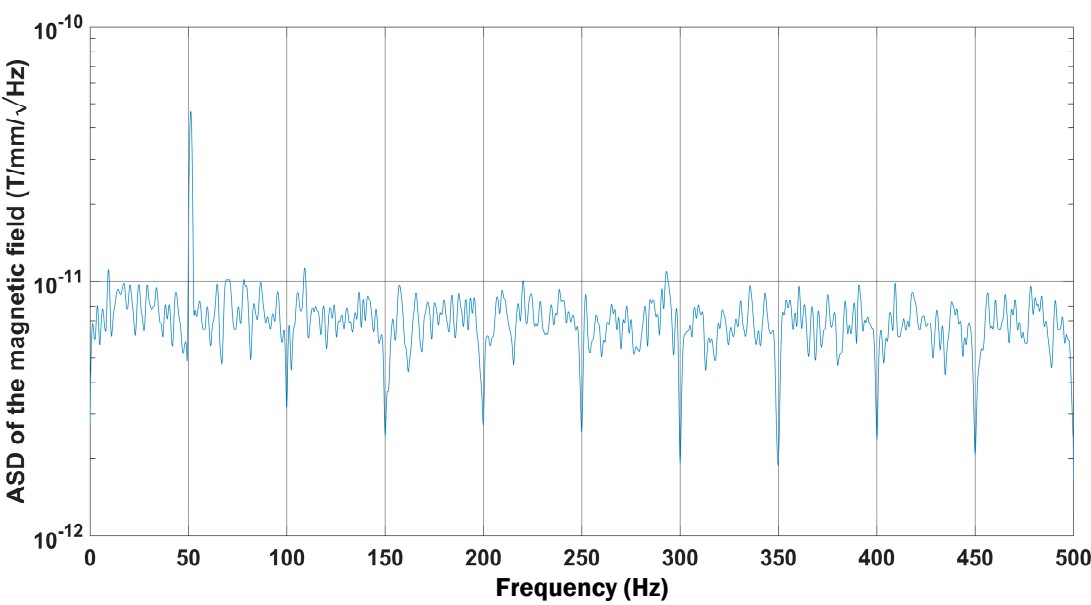

**Figure 7.** ASD of a pulsed magnetometer in gradiometric configuration with two beams at 1.5 mm distance inside a single cell illuminated by an external cavity diode laser.

For higher signal-to-noise (SNR), a co-linear configuration of the pulsed magnetometer with a vapor cell inside the pump laser cavity is used, as Figure 8 shows. The higher SNR originates from the fact that the volume of the pump–probe interaction is larger. Besides the main elements shown in Figure 3, this setup includes a polarizing cube, special dual wavelength waveplates of $\lambda/4$ at 895 nm and $\lambda/2$ at 852 nm and a bandpass filter at 852 nm.

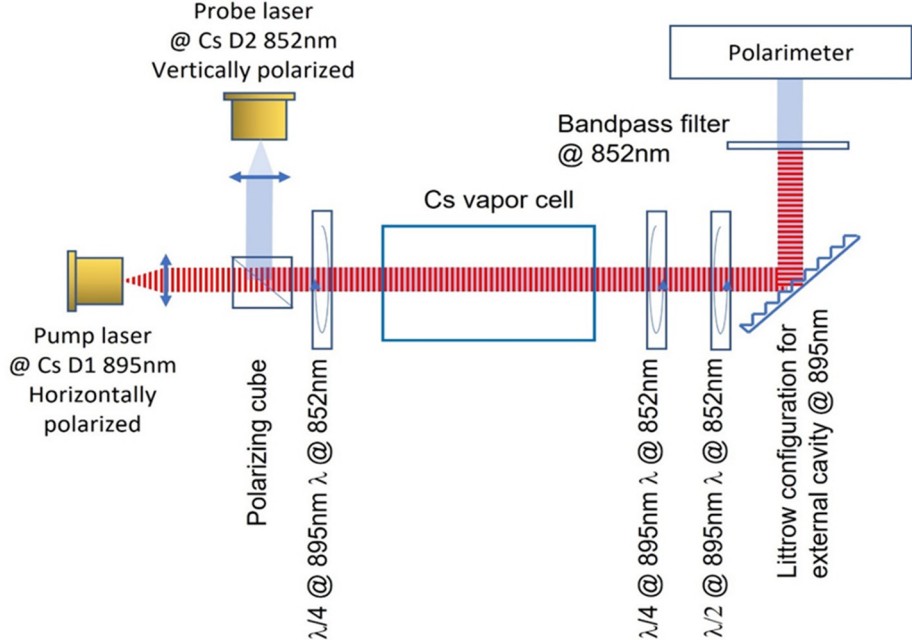

**Figure 8.** Pulsed magnetometer with co-linear laser beams and vapor cell inside the pump laser cavity.

## 4. Summary

A crucial novel step for realizing atomic correlations and quantum application of a vapor cell inside a lasing cavity is performed. This method can be extended toward focusing the coherent em field inside the laser cavity in quest of increasing the photon number that interacts with the atoms. Note that several optics elements related to the magnetometer

function are also introduced in the cavity which decreases n, the coherent photon number exposing the atoms. Yet, the coherent em field at resonance is achieved experimentally.

The internal cavity reflections along with the relatively long cavity-narrowing laser linewidth make for efficient atomic pumping. A maximum magnetometer signal is obtained with two parallel aligned waveplates at 45° to the original vertical laser polarization. The highly reflected S polarization in the Littrow configuration increases the intensity of the circular-polarized laser between the waveplates inside the cell. The cell-in-cavity configuration increases atoms' polarization and increases signal-to-noise ratio. Higher pump laser power, AR-coated cell, as well as an AR-coated diode laser, may further improve the signal.

As seen in the theory, the correlation process is proportional to the field intensity. Thus, a required next step is essential increase of the coherent field. The effect of the cavity is in providing the length of time for exposing the atoms to the coherent field and for establishing atoms' correlation. Also, we propose a much better physics arena for realizing the quantum application: cold atom devices. One possible way is to employ ultra short lasers that may provide the necessary strong coherent field to achieve atoms' correlation. We hope this article will encourage such research with its successful first-step operation of a cell-in-cavity laser achieved toward experimental quantum applications and demonstrated in an optical magnetometer.

**Author Contributions:** Conceptualization, R.S. and A.B.A.B.; Methodology, R.S. and A.B.A.B.; Software, G.A.K. and D.L.; Validation, A.B.A.B. and R.S.; Formal analysis, G.A.K., D.L. and R.S.; Investigation, A.B.A.B. and G.B.; Resources, R.S.; Data curation, A.B.A.B., G.A.K. and D.L.; Writing—A.B.A.B., G.A.K. and R.S.; Writing—review and editing, A.B.A.B.; Visualization, A.B.A.B.; Supervision, R.S.; Project administration, R.S.; Funding acquisition, R.S. All authors have read and agreed to the published version of the manuscript.

**Funding:** Ben-Gurion University of the Negev, internal funding. This research received no external funding.

**Institutional Review Board Statement:** Not applicable.

**Informed Consent Statement:** Not applicable.

**Data Availability Statement:** Data are available upon request.

**Conflicts of Interest:** The authors declare no conflicts of interest.

**Appendix A**

The following is a detailed presentation of our Toptica diode laser free-running linewidth and its linewidth after its narrowing by the Littrow configuration external cavity. Presented also are our pulse magnetometer definitions of properties: bandwidth, dynamic range and sensitivity.

1. Figure A1 shows the emission spectrum of Toptica diode laser LD-0860-0080-AR-1 (https://www.toptica.com/fileadmin/Editors_English/14_stocklists/AR-Stock-list.pdf accessed on 28 December 2023). The calibration is 2.35 nm/ms, therefore, the FWHM is approx. 5 nm. The diode is AR coated so there is no front mirror. The tuning range is 840–875. We narrowed and tuned the laser to resonance in the Littrow configuration external cavity. Figure A2 shows the narrowed laser spectrum. The measured linewidth by spectrometer is approx. 0.15 nm which is the spectrometer's resolution. The real linewidth was therefore measured by the wavemeter and is less than 0.2 pm (less than 60 MHz). For efficient optical pumping the pump-laser linewidth should be in the order of the atomic linewidth and certainly not much smaller to avoid hole burning. The linewidth of the probe laser is similar, and the wavelength is detuned from resonance by several natural linewidths (tens of picometer in our case) to avoid absorption and atomic polarization destruction.

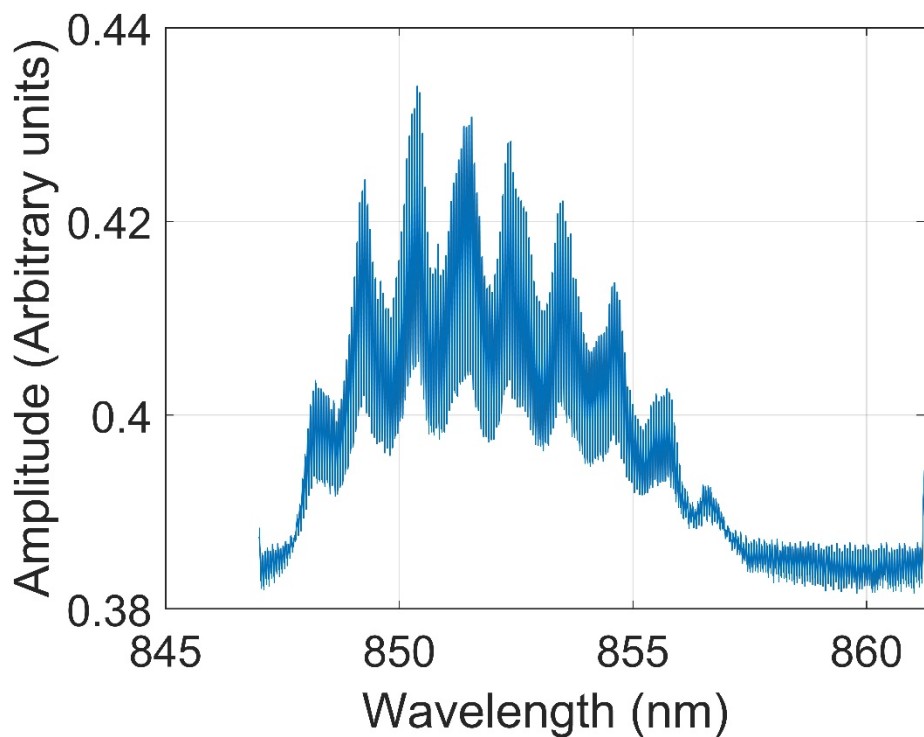

**Figure A1.** The emission spectrum of the Toptica AR-coated diode laser LD-0860-0080-AR-1 measured by a 30 cm spectrometer and recorded on a digital oscilloscope.

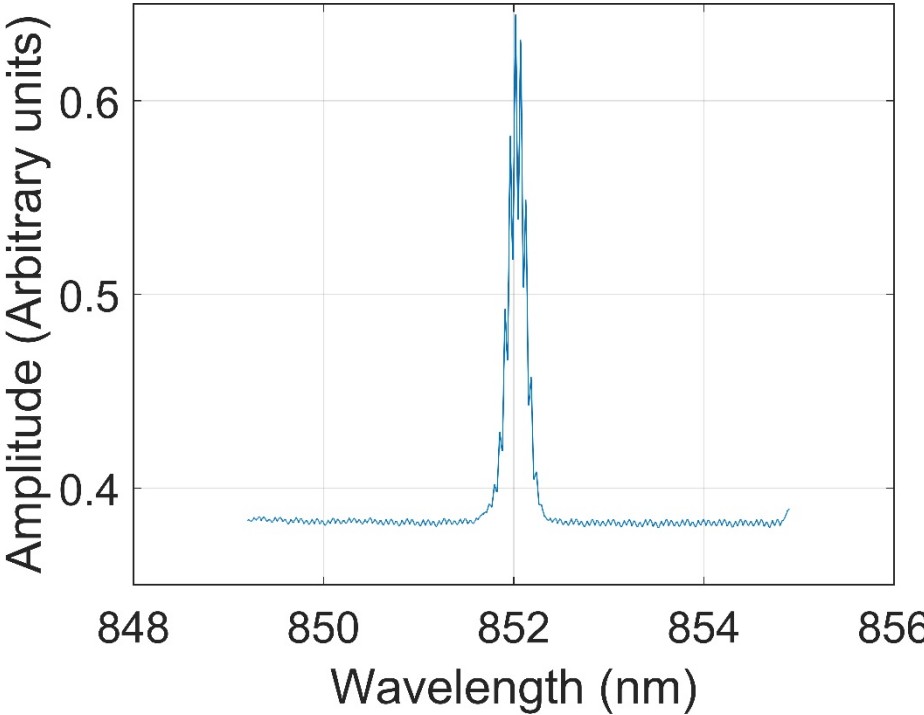

**Figure A2.** The emission spectrum of the Toptica AR-coated diode laser LD-0860-0080-AR-1 narrowed and tuned by a Littrow configuration external cavity. The linewidth is significantly smaller than the spectrometer's resolution and was separately measured by a wavemeter.

2. The linewidth is narrowed by the Littrow configuration external cavity and not by the Cs atoms or glass cell. Our claim is that, despite the absorption at resonance by Cs atoms, we locked the laser at the desired wavelength with a narrow linewidth,

adequate to optical pumping, i.e., in the same order as the natural atomic linewidth. With the cell in the cavity or without it, the linewidth is the same, defined only by the optical cavity.

3.  The maximum *bandwidth* of our pulsed magnetometer operated at 1 kHz is 500 Hz (as defined by Nyquist frequency), but the magnetometer is sensitive to much higher frequencies. However, its sensitivity decays with frequency and it reaches 3 dB decay at 225 Hz.

4.  The presented pulse magnetometer works unshielded in the earth's field whose amplitude changes with the latitude. The *dynamic range* is defined by the range of the ambient field for which minute temporal changes can be measured, and for our magnetometer it is 20–100 µT.

5.  The *sensitivity* of the magnetometer is the minimal measured intensity of the minute changes in ambient field, and it is better than $10 \text{ pT}/\sqrt{\text{Hz}}$. In our measurements, we applied various magnetic fields at different frequencies (up to 500 Hz), with intensities in the order of 1 nT. These fields are depicted on the background of Earth's field as shown in Figure 5.

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
