# Peer review of "Quantum Applications of an Atomic Ensemble Inside a Laser Cavity"

_photonics, doi:10.3390/photonics11010046_

Round 1
Reviewer 1 Report
Comments and Suggestions for Authors
Please see attached report.

Reviewer 2 Report
Comments and Suggestions for Authors
Please see the comments in the attached file.

Please see the comments in the attached file.
Author Response
Response to Reviewer 2
Thank you very much for your important comment.
We accepted reviewers’ comments and revised the paper accordingly. We added Appendix A1 for clarifications.
I agree with the reviewers about the novelty. This paper is intended to bring about a method to achieve atoms correlation. Achieving this is of major importance. We did not claim to achieve this effect, and rather suggested a method to realize it and to encourage other researchers, mainly in the cold atoms arena to try accomplishing it. Admittedly, atoms correlation is quite hard to achieve, and we are only at the very start.
A possible method to achieve atoms correlation is to employ ultra short lasers that may provide the necessary strong field. A calculations (complex) are underway.
Appendix1 is added for more clarifications.
Round 2
Reviewer 1 Report
Comments and Suggestions for Authors
Please see attached report.

Author Response
Thank you very much.
Attached are our responds.

Reviewer 2 Report
Comments and Suggestions for Authors
I have no further comments. I think this work can be accept in the present form.
Author Response
Thank you for your recommendations.